# Cyclic Federated Learning Method Based on Distribution Information Sharing and Knowledge Distillation for Medical Data

Liang Yu [1,2] and Jianjun Huang [1,2,*]

1 College of Electronic and Information Engineering, Shenzhen University, Shenzhen 518060, China
2 Guangdong Key Laboratory of Intelligent Information Processing, Shenzhen 518061, China
* Correspondence: huangjin@szu.edu.cn

**Abstract:** Federated learning has been attracting increasing amounts of attention for its potential applications in disease diagnosis within the medical field due to privacy preservation and its ability to solve data silo problems. However, the inconsistent distributions of client-side data significantly degrade the performance of traditional federated learning. To eliminate the adverse effects of non-IID problems on federated learning performance on multiple medical institution datasets, this paper proposes a cyclic federated learning method based on distribution information sharing and knowledge distillation for medical data (CFL_DS_KD). The method is divided into two main phases. The first stage is an offline preparation process in which all clients train a generator model on local datasets and pass the generator to neighbouring clients to generate virtual shared data. The second stage is an online process that can also be mainly divided into two steps. The first step is a knowledge distillation learning process in which all clients first initialise the task model on the local datasets and share it with neighbouring clients. The clients then use the shared task model to guide the updating of their local task models on the virtual shared data. The second step simply re-updates the task model on the local datasets again and shares it with neighbouring clients. Our experiments on non-IID datasets demonstrated the superior performance of our proposed method compared to existing federated learning algorithms.

**Keywords:** cyclic federated learning; non-IID; distribution information sharing; knowledge distillation



## 1. Introduction

Deep learning is widely used in clinical scenarios, such as disease screening, health management, diagnosis and treatment. Obtaining models that can perform various medical tasks well often requires a large amount of training data; however, due to privacy limitations in the medical field, it is not possible to pool data from various medical sites to form larger datasets, which isolates each medical site and means that models can only be trained with a small amount of local data, resulting in the poor performance of trained models. Federated learning [1] has been proposed as an effective solution to this problem. Firstly, as a kind of distributed machine learning, federated learning can jointly train global models for multiple medical institutions by combining data and annotations from each institution to expand the sample data volume and the number of annotations [2], thereby making it possible to solve unbalanced data distributions. Secondly, federated learning does not require data exchanges among healthcare institutions, which satisfies requirements such as patient privacy protection, data security and government regulations. Additionally, the results of federated learning can be shared among medical institutions, which can alleviate the problem of the uneven distribution of medical resources to a certain extent.

The training process of federated learning involves medical institutions training model parameters based on local datasets, then sharing model parameters among medical institutions and finally fusing all model parameters in an aggregated manner to form better-

performing models. When the data distributions of medical institutions are inconsistent, i.e., the assumption of independent and identical distribution (IID) is not satisfied among medical institutions, the complexity of the problem modelling, theoretical analysis and empirical evaluation of solutions increases, resulting in the degradation of model performance [3]. A feasible idea to solve this problem is to share data distributions based on the model sharing in federated learning, i.e., share the data distribution information of different medical institutions with other medical institutions. This is similar to the sharing and exchange of treatment experiences among doctors at multiple medical institutions, which can improve treatment levels by learning from each other. In addition, there are certain requirements for data security while keeping shared data.

The initial federated learning framework was the centralised federated learning framework, which faced the problem that it is difficult to find trusted third parties to perform parameter aggregation [4]. To solve this problem, decentralised federated learning frameworks have been developed, such as peer-to-peer network structures; however, they have certain requirements for the computing power of each client. Due to the frequent information exchanges between multiple clients, the communication costs are also relatively high. The decentralised federated learning architectures remove the central server to perform task model aggregation locally and only exchange information between adjacent clients on the communication graph, which reduces the probability of network congestion and communication overheads while improving data privacy protection capabilities. Therefore, these architectures are very suitable for the model exchange framework of federated learning and the exchange of shared data. Thus, based on this, our approach is proposed to improve the task model performance of federated learning for non-IID data.

To sum up, the main contributions of this work can be summarised as follows:

1. A novel unidirectional synchronous cyclic decentralised federated learning framework and an effective evaluation of the convergence of the model;
2. A new distribution information sharing and knowledge distillation model aggregation algorithm for the federated task model, which solves the problem of data distribution inconsistency both at the algorithm level and the data level;
3. The first attempt to use federated learning to diagnose Alzheimer's disease based on medical datasets;
4. A way to measure the inconsistent distributions of data features using the maximum mean difference (MMD).

The rest of our paper is organised as follows. Section 2 introduces related work. Section 3 details our proposed approach. Section 4 describes the experimental environment and our experimental results. Section 5 concludes the paper and proposes future work.

## 2. Related Work

Since federated learning was first proposed, four main types of challenges have arisen: communication challenges, system challenges, statistical challenges and privacy challenges [4]. We can refer to these two articles [5,6] for the communication challenges and system challenges of a cyclic federated learning framework, which have been analyzed and solved by predecessors. For privacy challenges, we can refer to the solutions in these two articles [7,8]. The privacy security protection strategies proposed in both papers consist of a privacy protection module and an attack detection module, while the major difference between the two is that the first scheme uses a two-level privacy data protection module. This scheme uses perturbation-based privacy converts categorical values into numeric and normalizes feature values into a range of [0, 1] before transforming the data using DL-based encoder techniques, which strengthens privacy and increases the utility of DL models .The statistical challenges, e.g., the non-independent and identical distribution of data (non-IID) problem, are some of the most non-negligible challenges in the application of federated learning in the medical field. Therefore, in this paper, we mainly focus on the non-IID problems.In response to non-IID problems, existing research has mainly solved the problems at the algorithm and data levels.

The algorithm-level solutions mainly include objective function modification and solution mode optimisation. Objective function modification involves adding regularisation terms on the client side. A trade-off has been achieved between optimising local models and reducing the differences between local models and global models to solve the non-independent homogeneous distribution of data at each node [9–12]. The measure of the differences between local models and global models by the regularisation terms can be either the distance between them or the differences in model behaviour. The distance measures between local and global models are Euclidean distances [9] and weighted distances [10]. For example, the federated proximal optimisation (FedProx) algorithm that has been proposed in the literature [9] corrects the client-side drift that occurs in FedAvg by restricting the Euclidean distances between local models and global models as proximal terms. This means that the local updates do not excessively deviate from the global models, which alleviates any inconsistencies in the client-side data and improves the stability of global model convergence. The federated curvature (FedCurv) algorithm that has been proposed in the literature [10] uses Fisher information from global models obtained during the previous rounds of training to weight the distances, which can reduce excessive errors in the model parameters. The differences in model behaviour between local and global models can be measured by the degree of inconsistency in the model output distributions on local datasets or by the gradient of the global models on local datasets. For example, in the literature [11], the maximum mean discrepancy (MMD) has been used as a metric to measure the inconsistency in model output distributions on local datasets. The stochastic controlled averaging (SCAFFOLD) algorithm that has been proposed in the literature [12] improves the FedProx algorithm by adding a control variable on the client side. This control variable can take either the gradient norm of global models on local datasets or the Euclidean distances between local and global models, thus preventing local models from deviating from the globally correct training direction. These methods can improve the performance of federated learning for model learning on non-IID datasets to some extent, but the degree of improvement is limited by the consistency of the client-side data sampling [3].

In solution optimisation, the good performance of federated learning models is mainly achieved by improving the server-side aggregation method. The ideal application conditions for federated learning are IID-based datasets (such as the initially proposed FedAvg algorithm) and weights for clients that are proportional to the number of samples. The accuracy of global models is greatly degraded in the case of the inconsistent, unbalanced and non-independent distribution of client data [13]. For this reason, most scholars have aimed to improve the shortcomings of aggregation methods for federated averaging algorithms. Accuracy-based averaging (ABAvg) has been in the literature [14], in which the server-side tests the accuracy of temporary models on validation datasets to obtain the accuracy of the models on the client side and then normalises them before aggregating all parameters. The federated learning with matched averaging (FedMA) algorithm that has been proposed in the literature [15] uses Bayesian non-parametric methods to match and average weights in a hierarchical manner. The federated averaging with momentum (FedAvgM) algorithm that has been proposed in the literature [16] applies momentum when updating global models on a server. The federated normalised averaging (FedNova) algorithm that has been proposed in the literature [17] normalises local updates before averaging. However, these methods have limited success in improving the performance of global models [12], so some scholars have proposed approaches that evade this problem, such as personalised federated learning, multitask federated learning and federated meta-learning, which can also improve the performance of federated learning on non-IID data to some extent.

The source of global model performance degradation is the non-IID problem; thus, data-level approaches to sharing client-side data have become new options for solving the non-IID problem. Client-side data sharing can be divided into two types: direct data sharing and indirect data sharing. In terms of direct data sharing for federated learning, one approach is to use a global sharing strategy [18–20], in which the server-side shares small

amounts of public data with the client for training to reduce the variance between trained local models, thus increasing the robustness and stability of the training process. This sharing approach relies on task-specific public datasets, and, in practice, there is a risk of privacy violation during both the acquisition and sharing of public data. Another approach is to use a local sharing strategy [21,22], in which small amounts of data are shared directly through trusted communication links between clients; however, this approach also violates the privacy preservation conventions of federated learning.

Indirectly shared federated learning does not share data directly, but rather makes the distributions of client datasets consistent by sharing data distribution information on the client side and then augmenting local training datasets with the shared distribution information [23,24]. The data distribution information can be learned using generator networks, which can be divided into global and local generators, depending on how the generators are trained. For example, a global generator shared approach has been proposed in the literature [23] that trains conditional generative adversarial network(CGAN) [25] generators on central servers and then shares the generators with clients to share distribution information. However, the data required for training CGANs using central servers are extracted from all clients, and there is a risk of privacy violation during the transmission of extracted data from the clients to the server side. A local generator shared approach has also been proposed in the literature [24] that trains bulldozer distance-based generative adversarial networks (i.e., Wasserstein generative adversarial networks, WGANs) [23] on local datasets on the client side and shares them with other clients. An image translation network is then trained using local generators and other generators to solve the federated learning problem for client-side heterogeneous data. Implicit data sharing through generators does not cause any privacy problems and is more practical than direct data sharing because it meets the need for patient privacy protection in healthcare organisations.

The data-immobile and model-immobile nature of federated learning has led to its increasingly widespread application in fields with high requirements for sensitive data protection, such as medicine. To address the problem of the degradation of federal learning performance due to inconsistent data distributions among federated learning participants, federated learning for client-side data sharing has become an effective solution strategy. Among the different options, the approach of sharing data distributions rather than the data themselves is more appropriate for application because it does not create the risk of privacy violation. Therefore, we addressed this issue by integrating solutions at both the data and algorithm levels. See Figure 1 for details of classification guidelines.

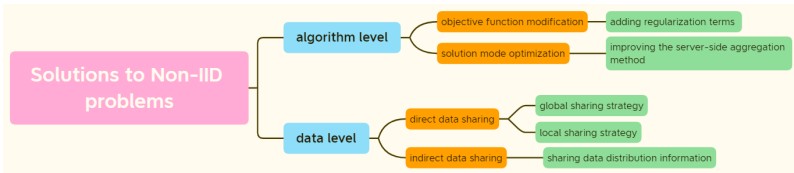

**Figure 1.** Solutions to non-IID problems.

### 3. The Distribution Information Sharing- and Knowledge Distillation-Based Cyclic Federated Learning Method

The ultimate goal of federated learning is to jointly train optimal models for multiple clients; in this paper, we refer to these as task models, which are made by multiple medical institutions to obtain target models. Task models can be for the diagnosis of diseases, lesion segmentation, etc. In federated learning, local task models tend to be consistent with global task models; however, in the case of non-IID local client data, local task models deviate from global task models. In the existing state-of-the-art circular decentralised federated learning schemes, the model parameters of nodes are updated after multiple steps of weighted summation and then averaged, which is a complex and costly communication strategy. In addition, the weighted average approach to model parameter aggregation often yields poor task model performance on non-IID datasets because the client data

distributions of neighbouring nodes may differ significantly and thus, the trained task models are biased. To address this, a natural idea is to degrade this bias by sharing data distributions to generate augmented datasets while preserving data privacy and then using the augmented data to learn the data distributions of other clients to achieve the implicit aggregation of model parameters. For this purpose, we used generators to learn the data distribution information of clients and share the local task models of clients, together with the local data generators, with neighbouring clients. Since both the generators and the task models carrying the data distribution information of the neighbouring clients were trained on the same datasets, this facilitated the use of the migration learning idea to aggregate the task models of two neighbouring clients. Based on this, we proposed a teacher–student model-based migratory learning approach for task model aggregation. Figure 2 shows a general block diagram of our proposed approach.

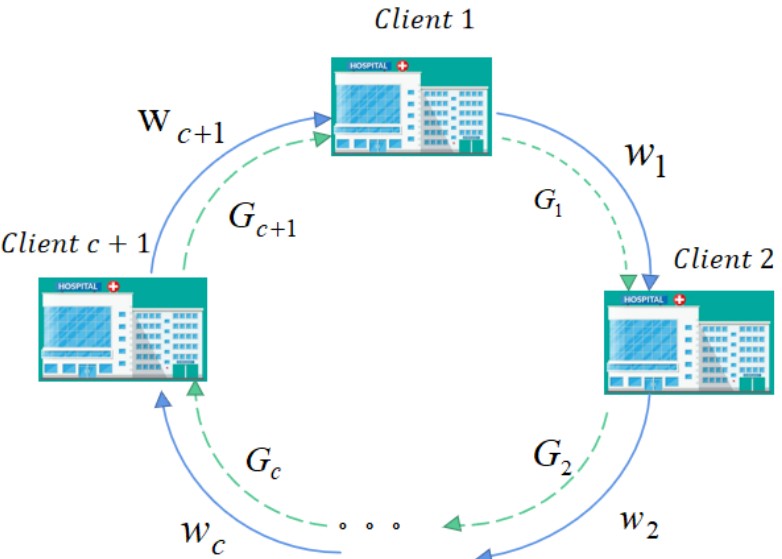

**Figure 2.** A schematic diagram of our cyclic federated learning method based on distribution information sharing and knowledge distillation.

Supposing that there are $C$ clients involved in the federated learning task (where $G$ is the shared generator model parameters that are locally trained offline, and $w$ is the task model parameters that are dynamic shared weights), the overall process can be divided into two stages as follows:

Stage 1: The offline process. All clients participating in the federated learning task train the generator network offline on local datasets to obtain the generator network $G$ that responds to local distribution information. Then, all clients pass the trained generator $G$ to the next client in turn. The next client $c+1$ generates the corresponding virtually shared local data after receiving the generator from client $c$ before.

Stage 2: The online process, which can be mainly divided into two steps. The first step is the knowledge distillation learning process, in which all clients first initialise the task model on local datasets and share it with the next client, and the next client then uses the shared task model to teach its task model on the data that were virtually shared via knowledge distillation. The second step simply re-updates the trained task model on local datasets again and shares it with the next client.

### 3.1. Distribution Information Acquisition Based on Deep Learning

To eliminate the adverse effects of the non-IID problem on the performance of medical institution federated learning, an effective approach is to augment the local datasets of medical institutions by sharing their data distributions. To obtain information about the data distributions of healthcare institutions, the current state-of-the-art approach is to use

a generator model with deep learning. Generators are the most effective tools for data augmentation because they not only learn the distribution information of data effectively but also generate data that match the real distributions. Generative adversarial networks (GANs), as one of the current types of mainstream deep neural network generators, are powerful in terms of image enhancement and image-to-image conversion [22]. Therefore, we adopted a GAN as a data generator on the main server to obtain the data distribution information of local clients [26–28] and added conditional information to generate the type of data that we needed, i.e., the final generator model was a CGAN. Specifically, let the total number of clients (federated learning participants) participating in the federated learning task be $C$, let the local datasets of the $c$ ($c = 1, 2, \cdots, C$) client be $D_c = \{x_i \mid i = 1, 2, \cdots, N_c\}$ and let $N_c = |D_c|$ be the number of clients in the training sample. The client $c$ trains a generator and reflects the distribution information $G_c$ of local datasets $D_c$. Thus, $C$ clients are trained to obtain $C$ generator models. The distribution of information obtained in this way is relatively safe from privacy breaches.

### 3.2. Distribution Information Sharing

The purpose of sharing distribution information is to enable later clients in the cyclic communication graph to have virtually shared data about the previous client's data distribution information, thus enabling two adjacent clients to achieve a consistent distribution of data to improve the performance of task models. To this end, we combined the features of a cyclic federated learning architecture and model parameters to accomplish this process. Let $c = 1, 2, \cdots, C$ and let the client $c$ transmit the generator $G_c$ to the client $c+1$. When $c = C$, let $c + 1 = 1$, thus forming a ring-shaped communication link. Under the condition of this cyclic communication link, let the client $c + 1$ receive the generator $G_c$ from the client $c$, where $N_c = |D_c|$ is the number of local data points from the client $c$. Accordingly, $G_c$ can generate $N'_{c+1}$ virtually shared data points, i.e., $D'_{c+1} = \{x_l \mid x_l = G(z_l), l = 1, 2, \cdots, N'_{c+1}\}$. Therefore, only the client $c+1$ has the distribution information of the client $c$, which indirectly realises distribution information sharing while protecting patient privacy. The distribution information sharing process is schematically illustrated in Figure 3.

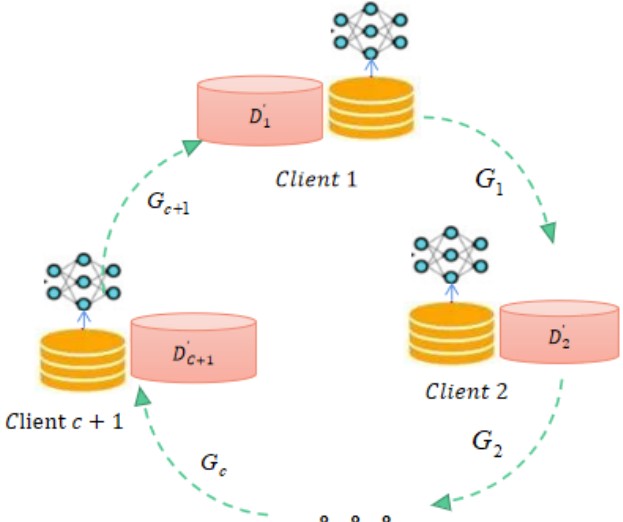

**Figure 3.** A schematic diagram of the distribution information sharing process.

### 3.3. Task Model Parameter Aggregation

The task model parameter aggregation process focuses on how to use shared distribution information for model parameter aggregation to eliminate the adverse effects of the non-IID problem on federated learning performance. In our cyclic federated learning framework, the client *c+1* not only receives the task model parameters from the client *c*

through a trusted channel but also the generator model $G_c$. The virtually shared data $D'_{c+1}$ can be generated locally via $G_c$. Since $D'_{c+1}$ have consistent distributions across the local datasets $D_c$ of the client $c$, the task model $f(x, w_c)$ obtained by the client $c$ after training using $D_c$ has a good performance. However, the distributions of the local datasets $D_{c+1}$ of the client *c+1* are usually not consistent with those of $D_c$, such that $f(x, w_c)$ performs worse on the local datasets $D_{c+1}$ of the client *c+1* than on $D'_{c+1}$. As a result, existing model aggregation algorithms, such as federated averaging and its various improvements, performed poorly in our cyclic federated learning framework. To this end, we proposed a new method for model aggregation for federation learning tasks based on knowledge distillation.

Since the locally trained task model of client $c$ has a similar optimal performance on datasets $D'_{c+1}$ and $D_c$, the locally trained task model $f(x, w_{c+1})$ of client *c+1* can be trained using the local task model $f(x, w_c)$ of client $c$ on the datasets $D'_{c+1}$ to improve performance. This idea could be implemented using the teacher–student model for migration learning, as shown in Figures 4 and 5.

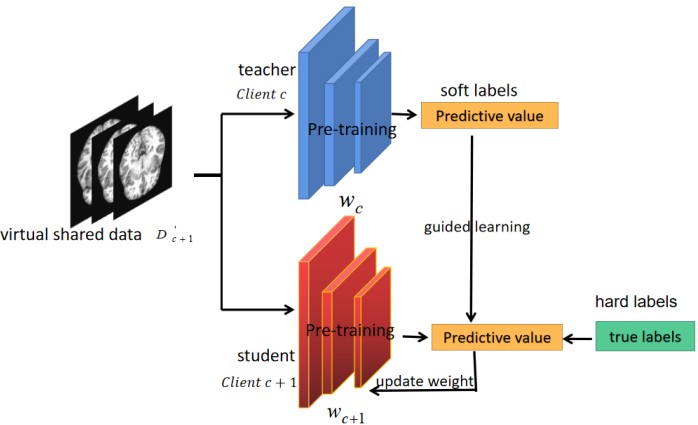

**Figure 4.** A schematic diagram of the teacher–student guided learning approach.

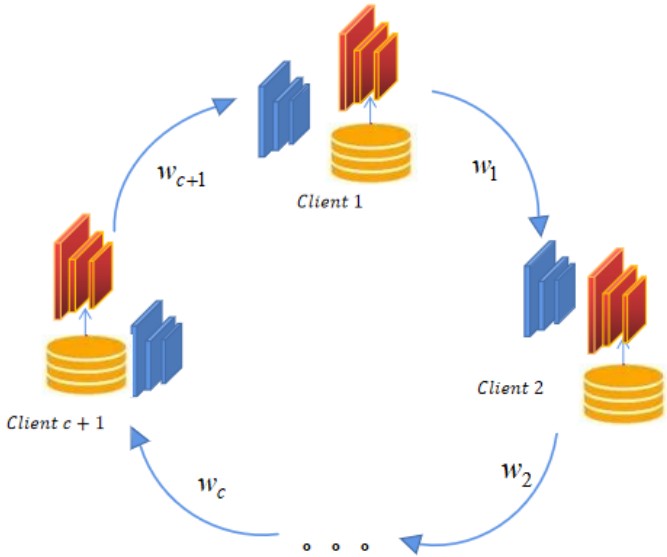

**Figure 5.** A schematic diagram of the teacher–student guided learning approach.

The training goal of our cyclic federated learning method based on the distribution of information sharing and knowledge distillation was the minimisation of the total loss function:

$$\ell(w_1, w_2, \cdots, w_C) = \sum_{c=1}^{C} L_{c+1}(w_{c+1}) + \lambda \sum_{c=1}^{C} R_{c+1}(w_{c+1}, w_c) \tag{1}$$

where $\gamma$ is a hyperparameter that controls the propensity of the local task model, $w_{c+1}$ is the parameter of the local task model of the client $c + 1$ (the task model to be trained can be the same or different for each client), and the loss function corresponding to $L_{c+1}(w_{c+1})$ has the following definition:

$$L_{c+1}(w_{c+1}) = \sum_{x \in D_{c+1}} l_{c+1}(x; w_{c+1}) \tag{2}$$

where $l_{c+1}(x; w_{c+1})$ is the loss of the task model $f(x, w_{c+1})$ on the data sample **x** and $R_{c+1}(w_{c+1}, w_c) \geq 0$ is the difference between the models of the adjacent clients $c$ and $c+1$ in the cyclic communication graph, which is defined as follows:

$$R_{c+1}(w_{c+1}, w_c) = \alpha L_{soft}(w_{c+1}, w_c) + \beta L_{havd}(w_{c+1}) \tag{3}$$

where

$$L_{soft}(w_{c+1}, w_c) = \sum_{x \in D'_{c+1}} l_{soft}(x; w_{c+1}, w_c) \tag{4}$$

$$L_{\text{hard}}(w_{c+1}) = \sum_{x \in D'_{c+1}} l_{\text{hard}}(x; w_{c+1}) \tag{5}$$

are the knowledge distillation loss and student loss on the datasets $D'_{c+1}$, respectively (which are defined in the same way as in the standard teacher–student model), and $\alpha$ and $\beta$ are two hyperparameters with values of 0 when the adjacent client models are the same and values of greater than 0 when they are different; the smaller the difference, the smaller the value (and vice versa). According to the incremental convex optimisation theory, the minimisation equation (Equation (1)) can be solved using the following iteration. At the $k$-th iteration, the gradient descent update is first performed on the intermediate variable $u_{c+1}$:

$$u_{c+1}^{(k)} = w_{c+1}^{(k-1)} - \alpha_k \nabla R_{c+1}\left(w_{c+1}^{(k-1)}, w_c^{(k-1)}\right) \tag{6}$$

where $\alpha_k$ is the gradient descent size, and the superscripts $k$ and $k+1$ denote the values of the $k$-th and $k$-th$+1$ iterations, respectively. Then, the model parameters are updated as follows:

$$w_{c+1}^{(k)} = \arg\min_{w} L_{c+1}(w) + \frac{\lambda}{2\alpha_k}\left\| w - u_{c+1}^{(k)} \right\|^2 \tag{7}$$

Using Equation (6), the iteration of $u_{c+1}^{(k)}$ learns the behaviour of $f\left(x, w_c^{k-1}\right)$ on the datasets $D'_{c+1}$, thus optimising the performance of the local model $f\left(x, w_{c+1}^k\right)$ that was updated using Equation (7) on the datasets $D_{c+1} \cup D'_{c+1}$. After multiple further iterations of training, as shown in Figure 4, all clients can learn the features of the data distributions of other clients via this cyclic framework, i.e., the training effect of a global model is reached. Ultimately, the adverse effects of the non-IID problem on medical institution-federated learning performance can be eliminated.

The above solution process can be described in pseudo-code as shown in Algorithm 1.

---

**Algorithm 1** Federated learning algorithm based on distribution information sharing and knowledge distillation.

---

**Input:** $C$ clients,each with its own training datasets$D_c$,generator $G_c$ and its own task model
　　$f(\boldsymbol{x}, \boldsymbol{w_c})$
**Output:** Trained model parameter set $\{\boldsymbol{w}_1, \boldsymbol{w}_2, \cdots, \boldsymbol{w}_C\}$

 1: **for** $c = 1, 2, \cdots, C$ **do**
 2:　　Client $c$ sends $G_c$ to Client $c+1$
 3:　　Client $c+1$ generates virtual shared data $D'_{c+1}$ with $G_c$
 4: **end for**
 5: **for** $k = 1, 2, \cdots, K$ **do**
 6:　　**for** $c = 1, 2, \cdots, C$ **do**
 7:　　　　Client $c$ sends $\boldsymbol{w}_c^{(k-1)}$ to Client $c+1$
 8:　　　　Client $c+1$ updates $\boldsymbol{u}_{c+1}^{(k)}$ according to (3) and (6)
 9:　　　　Client $c+1$ updates $\boldsymbol{w}_{c+1}^{(k)}$ according to (1) and (7)
10:　　**end for**
11: **end for**

---

## 4. Experimental Results and Discussion

### 4.1. Development Environment and Datasets

Our machine learning model was built by the well-known deep learning framework PyTorch, version 1.6.0, and Python, version 3.7.1. A self-built cyclic federated learning framework was used, in which a Kafka cluster was used as the information medium for model parameter exchange. The generator network used six convolutional layers with a convolutional kernel size of 4x4.We used two datasets to validate the effectiveness of the proposed method, one of which was the Alzheimer's dataset that was used in a Kaggle competition .For this dataset, we used the pre-trained model VGG16 provided by torchvision as our classifier network. The second dataset was the MNIST dataset, which was also used to validate the generality of the proposed method, i.e., the generalisation ability of the method. For this dataset, we used the two-layer convolutional layer network used in MOON. The Alzheimer's dataset, which has a total of 5120 training data points and 1279 testing data points, has a 1:1 ratio of diseased to non-diseased data in both the testing and training sets. The MNIST dataset has a total of 60,000 images in the training set and 10,000 images in the testing set.

### 4.2. Experimental Parameters

There are various scenarios of non-IID data. In this study, we focused on two of them: attribute skew and label skew. To study these two different types of data distributions, we conducted experiments on the two selected datasets. For the Alzheimer's dataset, due to its high data latitude and the few types of labels, we used the maximum mean difference (MMD) to measure the attribute skew of the client data [29]. The maximum mean difference is mainly used to measure the distance between two data distributions. Given two data distributions, the square of their MMD can be expressed as:

$$MMD^2(\text{x}, y) = \|E[\varphi(x)] - E[\varphi(y)]\|^2 \tag{8}$$

where $\phi(\bullet)$ denotes the mapping to the regenerated Hilbert space (RKHS). The inconsistency of client data distributions was measured by calculating the MMD, and the entire dataset was divided according to the MMD value to measure the federated learning performance under different MMD values. As for the MNIST dataset, we used the Dirichlet distribution [30] to divide the non-IID samples because of the many types of labels. Figure 6 shows the Dirichlet distribution when $\alpha = 0.5$ and the number of clients was 10.

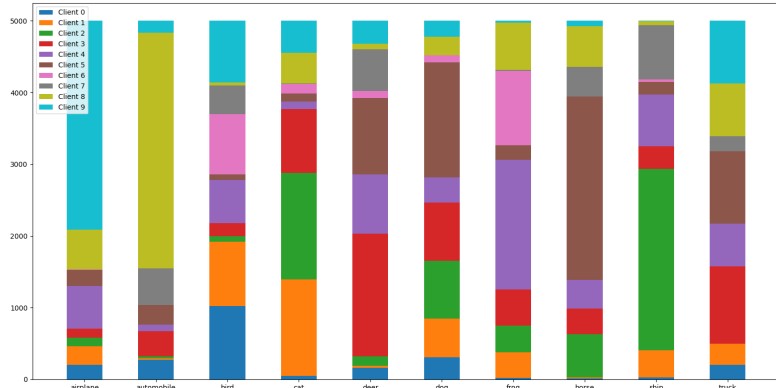

**Figure 6.** Dirichlet distribution.

*4.3. Algorithm Evaluation*

For the two distribution types (attribute skew and label skew), we effectively evaluated our algorithm on the selected datasets.

### 4.3.1. Attribute Skew

To study attribute skew, we conducted a correlation experiment on the Alzheimer's dataset, the results of which are shown in Table 1 and Figures 7–10. By comparing the accuracy rates on the testing set that are shown in Table 1, it can be seen that, in different MMD scenarios, the cyclic federated average-based method had a larger model loss, its performance was different from that of the centralised learning method, and the training was unstable. Our proposed method outperformed the cyclic federated average method, and the performance was close to or attained the centralised learning performance. The box line plot in Figure 10 shows that the MMD increased from the top left to the bottom right. By dynamically increasing the MMD, we could see that as the MMD increased (i.e., as the data distribution became more inconsistent), the model performance of the cyclic federated average method degraded faster and deviated greatly from the centralised learning performance, while the performance of our proposed method was better than that of the federated average, and the deviation from the centralised learning performance was slower.

**Table 1.** The top-1 accuracy of different MMD values on the Alzheimer's dataset.

| Data Division ID | MMD | CFL_DS_KD | CFL_FedAvg | CL |
|---|---|---|---|---|
| 1 | 0.514 | 79.95% | 78.97% | |
| 2 | 1.029 | 79.56% | 78.11% | |
| 3 | 1.283 | 78.73% | 77.24% | 79.22% |
| 4 | 1.546 | 78.60% | 72.20% | |
| 5 | 1.803 | 78.77% | 75.37% | |
| 6 | 2.059 | 78.05% | 70.00% | |

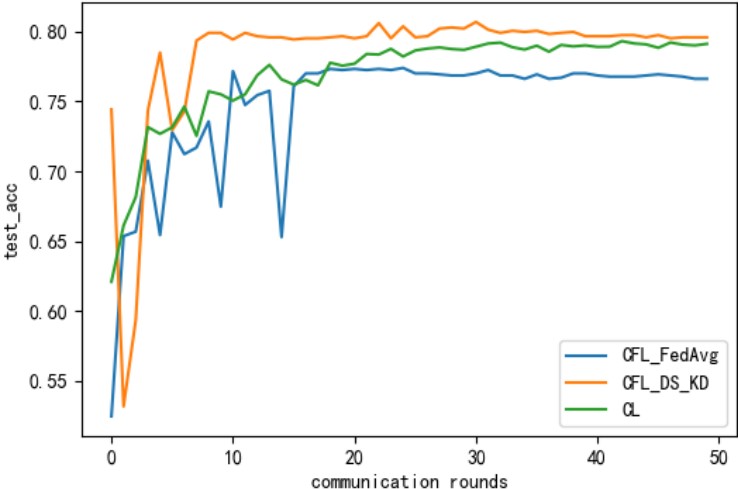

**Figure 7.** The accuracy of the different methods on the testing set after different amounts of communication rounds.

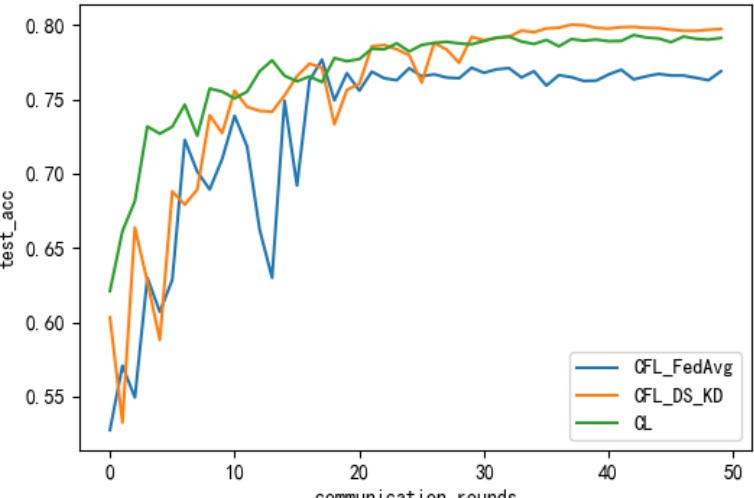

**Figure 8.** The accuracy of the different methods on the testing set after different amounts of communication rounds.

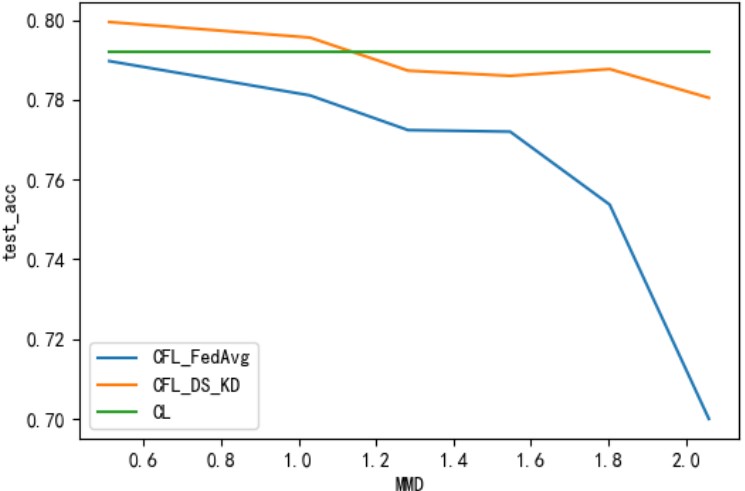

**Figure 9.** A performance comparison of the different methods under different MMD values.

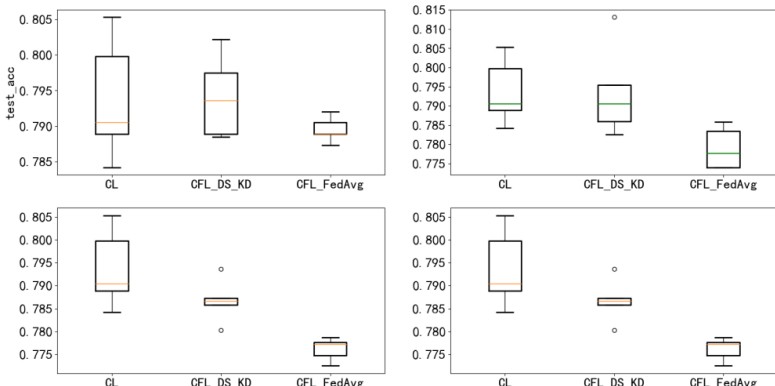

**Figure 10.** A performance comparison of the different methods under different MMD values.

Additionally, to further verify the effectiveness of our method, the output differences between the proposed method and the centralised learning method were analysed. When the difference was smaller, it meant that the proposed method was closer to the centralised learning performance. From Figure 11, it can be seen that the output performance difference between our method and the centralised learning method was almost 0, which effectively illustrated the beneficial effects of our method.

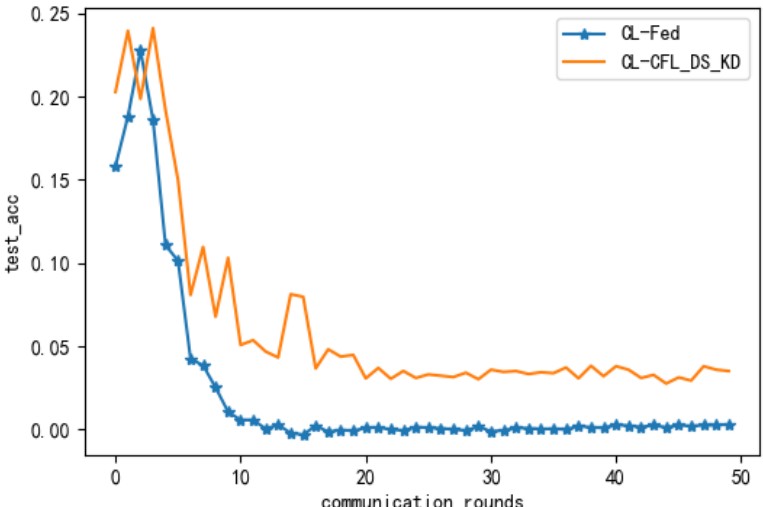

**Figure 11.** The difference between the CFL_DS_KD and CL method outputs.

### 4.3.2. Label Skew

The superior performance of our method was effectively demonstrated after several experiments on the Alzheimer's dataset. To further demonstrate the performance of our proposed method in the case of label skew, we also conducted corresponding comparative experiments on the public MNIST dataset. Comparisons were made between our proposed method and the state-of-the-art federated learning algorithm MOON and the mutual learning method Def_KT within a centreless federated learning framework. As shown in Figure 12 and Table 2, $\alpha$ is the Dirichlet distribution coefficient, and the smaller its value, the more inconsistent the data distribution. From the experimental results, it can be seen that the classification accuracy of our proposed method on the testing set was almost comparable to the centralised learning method and higher than those of MOON [31], Def_KT [32] and FedAvg. Thus, the superiority of our proposed method was effectively demonstrated, both on a medical dataset and a publicly available natural dataset.

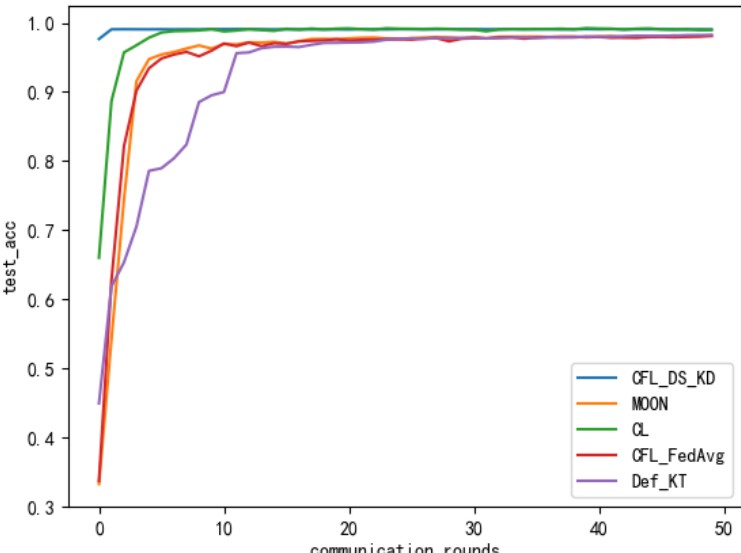

**Figure 12.** A performance comparison of the different methods when $\alpha = 0.5$.

**Table 2.** The top-1 test accuracy with $\alpha = 0.5$ and $\alpha = 0.1$ on MNIST datasets.

| Method | $\alpha = 0.1$ | $\alpha = 0.5$ |
| --- | --- | --- |
| MOON | 95.7% | 98.1% |
| CFL_FedAvg | 97.1% | 98.1% |
| Def_KT | 95.2% | 99.0% |
| CFL_DS_KT | 98.9% | 99.1% |

4.3.3. Convergence Evaluation

The convergence of centralised federated learning has been effectively proven, whereas that of centreless cyclic federated learning frameworks has not yet been proven. Therefore, in addition to the performance of the selected methods described above, we also experimentally evaluated the convergence of our cyclic federated learning architecture. We used two parametric numbers to find the differences in weights between clients. The weight differences could be expressed as follows:

$$D_l = \frac{1}{C} \sum_{c=1}^{C} \left\| w_{c+1}^l - w_c^l \right\|_2 \tag{9}$$

$$D_l^i = \frac{1}{C} \sum_{c=1}^{C} \left\| w_{c+1}^l i - w_c^l i \right\|_2 \tag{10}$$

Equation (9) represents single-layer weight differences, and Equation (10) represents single-weight differences in each layer, where $w_c^l$ denotes the $c$-th client's $i$-th layer weight and $w_c^{li}$ denotes the $i$-th weight difference in the $i$-th layer of the $c$-th client. We conducted experiments on the Alzheimer's dataset. Figure 13 shows the single-layer weight differences, in which it can be seen that as the amounts of communication rounds increased, the weight differences degraded sequentially and eventually stabilised. Figures 14 and 15 show the single-layer single-weight differences, in which it can be seen that the largest weight difference was in the thirteenth convolutional layer, and the maximum difference was only 0.0035 (i.e., close to 0) and reached convergence.

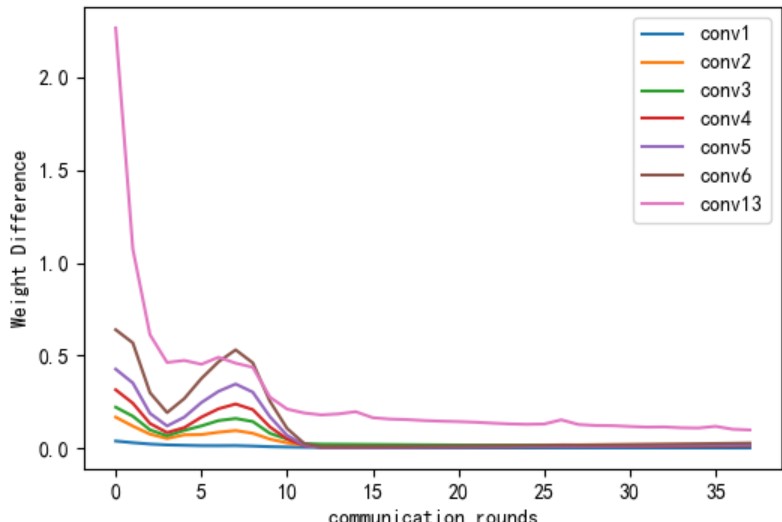

**Figure 13.** The single-layer weight differences.

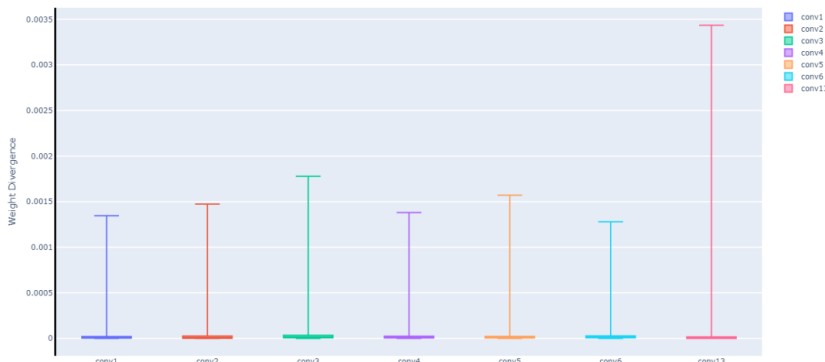

**Figure 14.** The single-weight differences in each layer.

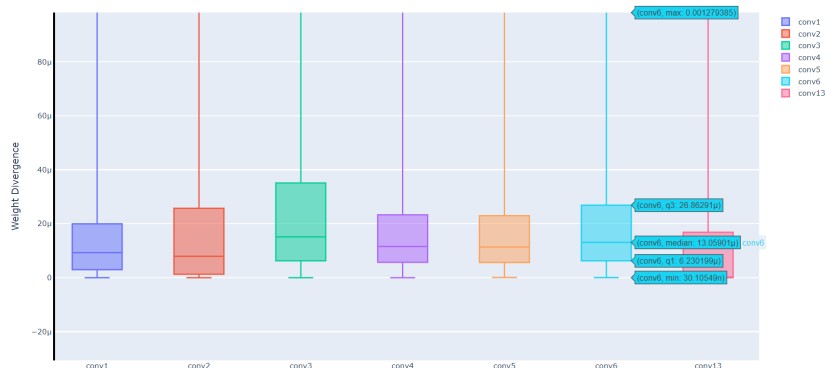

**Figure 15.** The single-weight differences in each layer.

The output variance could be expressed as follows:

$$DM = \frac{1}{C} \sum_{c=1}^{c} \| f(w_{c+1}; x) - f(w_c; x) \|_2 \tag{11}$$

where $f(w_c; x)$ denotes the output of the model of the $c$-th client under the input sample $x$. The output difference results are shown in Figure 16, in which the diagnosis difference represents the differences between the diagnosis results and the input data. The disease

output difference referred to the output differences between the results that were diagnosed as diseased, and the model output difference referred to the overall differences between the model outputs. It can be seen from the results that the output differences were small on the whole, i.e., very close to 0. Therefore, from the above experiments, we could conclude that the convergence of our proposed method was effectively evaluated. At present, the model is convergent.

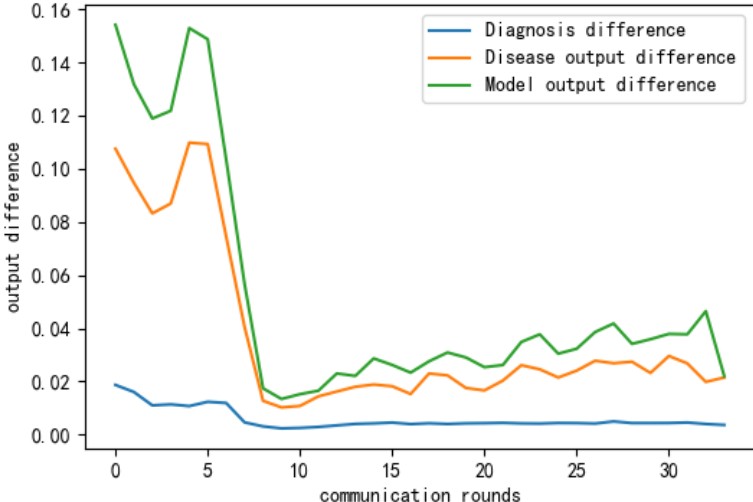

**Figure 16.** The output differences.

## 5. Summary

To address the non-IID problem in medical institution federated learning that cannot be effectively solved using existing federated learning techniques, this paper proposed a cyclic federated learning method (CFL_DS_KT) based on distribution information sharing and knowledge distillation. This is a novel and effective federated learning approach and, to the best of our knowledge, the first time we have used this unidirectional synchronous cyclic decentralised federated learning framework and effectively evaluated the convergence of a model with this structure. The experimental results also show that the task model achieves convergence under our proposed approach. Furthermore, in contrast to existing scholarly research solutions, we solve the non-IID problem by optimising the solution through the solution approach of distribution sharing and knowledge distillation. By considering both data-level and algorithm-level optimisation approaches, we achieve better performance of the federation learning model under non-IID while safeguarding client data privacy. In our extensive experiments on medical and public datasets, CFL_DS_KT shows a good improvement over various state-of-the-art methods, and its accuracy is closer to that of centralised learning. Further improvements in privacy preservation were achieved due to using a cyclic federated learning method. It also provided the idea of training federated learning models on heterogeneous data, which could eliminate data heterogeneity by transforming the data distribution information from one client to another.

However, our proposed approach has some shortcomings. When the client data is extremely heterogeneous, it is difficult to train a good generator to generate high-quality images due to the small amount of training data. Additionally, it is not suitable to train federated learning models with large numbers of clients as this could increase breakpoint failures and model training cycle times. Therefore, this method would mainly be suitable for federated learning across medical institutions.

**Author Contributions:** Conceptualization, L.Y. and J.H.; methodology, L.Y.; software, L.Y.; validation, L.Y. and J.H.; formal analysis, L.Y. and J.H.; investigation, L.Y.; resources, L.Y.; data curation, L.Y.; writing—original draft preparation, L.Y.; writing—review and editing, J.H.; visualization, L.Y.;

supervision, J.H.; project administration, J.H.; funding acquisition, J.H. All authors have read and agreed to the published version of the manuscript.

**Funding:** This research was funded by Science and Technology Planning Project of Shenzhen Municipality (No. 20200821152629001).

**Institutional Review Board Statement:** Not applicable.

**Informed Consent Statement:** Not applicable.

**Data Availability Statement:** Not applicable.

**Conflicts of Interest:** The authors declare no conflict of interest.

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
