# Peer review of "Cyclic Federated Learning Method Based on Distribution Information Sharing and Knowledge Distillation for Medical Data"

_electronics, doi:10.3390/electronics11234039_

Round 1
Reviewer 1 Report
This research topic is important to the next generation of networks, and the structure of the paper is good. However there are several small issues in the current manuscript:
Please improve the quality of Fig.3. Some words are too vague. Maybe the authors can increase the font size.
The authors should include the training time and testing time/instance for the adopted methodology and conduct the necessary analysis.
Et al. should be italicized and forward and backward quotations should be fixed
A reference link should be added to the tables and figures to enhance readability.
Attacks mentioned in "PEFL: Deep Privacy-Encoding-Based Federated Learning Framework for Smart Agriculture" and "DLTIF: Deep learning-driven cyber threat intelligence modeling and identification framework in IoT-enabled maritime transportation systems" needs to be considered as security and privacy both are serious concern.
Reviewer 2 Report
This article proposes a cyclic federated learning method based on the distribution of information sharing and knowledge distillation. The authors show a very good knowledge of the subject area, which allows them to successfully propose a new approach that leads to greater productivity.
The research methodology is well presented, experiments are done and the results are correctly evaluated.
The improvements that can be made to the article are as follows:
- to improve the quality of Figure 1;
- to give the achieved results in the conclusion more clearly.
Round 2
Reviewer 1 Report
1. Consider adding a table to summarize the systems in the literature review/related work to improve readability.
2. Authors should add and show how their approach is different from "PEFL: Deep Privacy-Encoding-Based Federated Learning Framework for Smart Agriculture" and "DLTIF: Deep learning-driven cyber threat intelligence modeling and identification framework in IoT-enabled maritime transportation systems, Permissioned blockchain and deep-learning for secure and efficient data sharing in industrial healthcare systems. Authors should Include them in related study.
Round 3
Reviewer 1 Report
accept